# Sequence and Structure Properties Uncover the Natural Classification of Protein Complexes Formed by Intrinsically Disordered Proteins via Mutual Synergistic Folding

**DOI:** 10.3390/ijms20215460

**Published:** 2019-11-01

**Authors:** Bálint Mészáros, László Dobson, Erzsébet Fichó, István Simon

**Affiliations:** 1MTA-ELTE Momentum Bioinformatics Research Group, Department of Biochemistry, Eötvös Loránd University, Pázmány Péter stny 1/c, H-1117 Budapest, Hungary; 2European Molecular Biology Laboratory, Structural and Computational Biology Unit, Meyerhofstraße 1, 69117 Heidelberg, Germany; 3Protein Structure Research Group, Institute of Enzymology, RCNS, HAS, Magyar Tudósok krt 2, H-1117 Budapest, Hungary; ficho.erzsebet@ttk.mta.hu; 4Membrane Protein Bioinformatics Research Group, Institute of Enzymology, RCNS, HAS, Magyar Tudósok krt 2, H-1117 Budapest, Hungary; dobson.laszlo.imre@itk.ppke.hu; 5Faculty of Information Technology and Bionics, Pázmány Péter Catholic University, Práter u. 50A, H-1083 Budapest, Hungary

**Keywords:** intrinsically disordered protein, IDP, protein–protein interaction, mutual synergistic folding, coupled folding and binding, structural analysis, structure-based classification, fold recognition

## Abstract

Intrinsically disordered proteins mediate crucial biological functions through their interactions with other proteins. Mutual synergistic folding (MSF) occurs when all interacting proteins are disordered, folding into a stable structure in the course of the complex formation. In these cases, the folding and binding processes occur in parallel, lending the resulting structures uniquely heterogeneous features. Currently there are no dedicated classification approaches that take into account the particular biological and biophysical properties of MSF complexes. Here, we present a scalable clustering-based classification scheme, built on redundancy-filtered features that describe the sequence and structure properties of the complexes and the role of the interaction, which is directly responsible for structure formation. Using this approach, we define six major types of MSF complexes, corresponding to biologically meaningful groups. Hence, the presented method also shows that differences in binding strength, subcellular localization, and regulation are encoded in the sequence and structural properties of proteins. While current protein structure classification methods can also handle complex structures, we show that the developed scheme is fundamentally different, and since it takes into account defining features of MSF complexes, it serves as a better representation of structures arising through this specific interaction mode.

## 1. Introduction

Intrinsically disordered proteins (IDPs) are crucial elements of the molecular machinery indispensable for complex life [1,2]. IDPs are parts of regulatory pathways [3], control the cell cycle [4,5], function as chaperones [6,7], and regulate protein degradation [8,9], amongst other functions. In accord, IDPs are typically under tight regulation at several levels [3,10]. While some IDPs fulfill their functions directly through their lack of structure, such as spring-like entropic chains, the majority of disordered proteins interact with other macromolecules, most often other proteins [11]. IDP-mediated interactions are essential for many hub proteins [12,13], and several IDPs serve as interaction scaffolds/platforms for macromolecular assembly [14,15]. Mounting evidence also shows that protein disorder plays a crucial role in the assembly of liquid–liquid phase separated non-membrane-bounded organelles [16].

Depending on the partner protein and the specifics of the interaction, IDPs can bind through several mechanisms. Several IDPs recognize and bind to ordered protein domains, usually through a linear sequence motif [17]. While some IDPs retain their inherent flexibility in the bound form as well [18], in most known cases the complex structure lends itself to standard structure determination methods, such as X-ray crystallography or NMR. These cases of coupled folding and binding have been studied intensively [19,20,21]. However, IDPs can utilize a fundamentally different molecular mechanism for interaction, through which they reach a folded state as well. Complexes that contain only IDPs as constituent protein chains, without the presence of a previously folded domain, are formed via a process called mutual synergistic folding (MSF) [22]—a much less understood way in which protein folding and binding can merge into a single biophysical process.

A major advancement in the field of IDP interactions in recent years was the development of specialized interaction databases for various mechanisms including coupled folding and binding [23,24], fuzzy complexes [25], mutual synergistic folding [26], and proteins driving liquid–liquid phase separation [27]. Out of these aspects, possibly the most understudied one is mutual synergistic folding, owing to the fact that these are the only interactions where none of the partner proteins have a well-defined structure outside of the complex, forcing us to revise our current approaches used for describing protein structures and complexes. The biological and biophysical properties of these interactions are markedly different from those mediated by other types of proteins. While in other interaction types a stable, folded hydrophobic core is already present in at least one partner, here the folding and binding happen at the same time for all partners. Comparative analysis has not only shown that MSF complexes constitute a separate biologically meaningful class, but also highlighted that these complexes are highly heterogeneous in terms of sequence and structure propreties [28,29,30].

We now have knowledge of over 140,000 protein structures deposited in the Protein Data Bank (PDB) [31], a major part of which contains several proteins in complex. In each of these cases, the proteins achieve stability either before or upon interacting. A major question is how is stability achieved? Can this be a basis of the definition of biologically meaningful classification? In the case of ordered proteins, current hierarchical classification schemes are rooted in the tertiary protein structures, such as in the case of methods/databases as SCOP (Structural Classification of Proteins) [32] and CATH (Class, Architecture, Topology, Homologous superfamily) [33]. While these methods are extended to classify protein complexes as well, they do not explicitly factor in parameters that describe the interactions or the differences in sequence composition between complexes of similar overall structures. However, in the case of MSF complexes, these differences are defining features, as the interaction is the primary reason for the emergence of the structure itself, and this interaction usually requires highly specialized residue compositions [28]. While other classification methods were developed specifically for protein–protein interactions, they only aim to describe the interface, without taking the overall resulting structure into account [34].

Here we present the first classification method designed to identify biologically relevant types of protein complexes formed via mutual synergistic folding. Our work aims to answer specific questions about the types of MSF complexes based on the currently known more than 200 examples. Are there intrinsic classes of MSF complexes or are all known examples basically unique in terms of sequence and structure? If meaningful groups are definable in an objective way, what are the characteristics of each group in terms of sequence composition and adopted structure? In addition, how is the formation of MSF complexes regulated? Are mechanisms known to be important for other molecular interactions relevant to these complexes as well? If so, are there differences between various MSF groups regarding these regulatory mechanisms and other biologically relevant properties, such as binding strength and subcellular localization?

## 2. Results

### 2.1. Sequence-Based Properties Define Four Clusters of Complexes

Complexes formed by mutual synergistic folding were taken from the MFIB (Mutual Folding Induced by Binding) database [26], and each complex has been assigned a feature vector describing the sequence composition of its constituent protein chains. To represent the sequence composition, we use the amino acid grouping previously used for investigating protein–protein complexes involving IDPs [28] (see Data and Methods and Figure 1 for definitions, and Appendix A for exact values for all complexes). These vectors were used as input for hierarchical clustering (Appendix A) to quantify the sequence-based relationship between various complexes. k-means clustering (Appendix A) indicates four as a suitable number of clusters, and, therefore, we use four sequence-based clusters in all subsequent analyses. While this choice is not the only acceptable one based on the k-means results, we aim to have a restricted set of clusters to describe the major types of sequential classes. The main features of the four clusters are shown in Figure 1, while cluster numbers for each complex are shown in Appendix A.

Figure 1 shows the average sequence compositions of each of the four sequence-based clusters. While clusters were defined based on sequence compositions only, Figure 1 also shows the average heterogeneity of the four clusters, meaning the average normalized difference in sequence composition between the interacting proteins of the complexes (see Data and Methods). Complexes in clusters 1 and 2 are both largely devoid of special residues, including Gly (flexible), Pro (rigid), and Cys (cysteine). Members of these two clusters contain an average fraction of hydrophobic residues; however are slightly depleted in aromatic residues, indicating that π–π interactions are not the dominant source of stability. The most characteristic difference between clusters 1 and 2 is that members of cluster 1 typically contain a high fraction of polar residues, while members of cluster 2 are enriched in charged residues. Also, cluster 1 members are typically formed by proteins with highly different compositions (high heterogeneity values), while cluster 2 members are formed by proteins of very similar compositions.

In contrast, members of clusters 3 and 4 are typically enriched in Gly and Pro and contain a higher-than-average fraction of aromatic residues. Again, polar/charged residue balance is a distinguishing feature, with clusters 3 and 4 showing preferences for polar and charged residues, respectively. Also, similarly to clusters 1 and 2, there is a notable difference in heterogeneity values between clusters 3 and 4: members of clusters 3 and 4 are typically composed of proteins with very similar and different residue compositions, respectively.

### 2.2. Structure-Based Properties Offer A Different Means of Defining Complex Types

The structural properties of the studied complexes were quantified using various features describing secondary structure compositions, various molecular surfaces, and incorporating hydrophobicity measures and atomic contacts (see Appendix A and Data and Methods). These structural features were used to describe each complex in the form of a feature vector, and similarly to the analysis of sequence properties, these vectors were input to hierarchical clustering; however, structural features were filtered, and only those that share a modest degree of correlation were kept (see Appendix A and Data and Methods for specifics) to avoid bias. The resulting tree is shown in Appendix A. In contrast to the sequence-based clustering, k-means within-cluster sum of squares analysis does not indicate any low number of clusters as more optimal than others (Appendix A). In order to have a medium number of clusters, we cut the hierarchical tree at a linkage distance that defines five clusters (Appendix A), again reflecting our preference to arrive at a moderate number of complex types, to provide a high-level classification scheme. The average values of structural parameters for all five structure classes are shown in Figure 2.

The obtained clusters show distinguishing structural features. Members of cluster 1 incorporate the highest amount of nonhelical secondary structure elements. These complexes heavily rely on a large number of buried hydrophobic residues for stability, and most stabilizing atomic contacts are formed between residues of the same protein, relying less on intermolecular interactions, which tend to be mostly polar in nature.

In contrast, members of cluster 2 adopt mainly helical structures. The stability of these complexes seems to rely more on the interactions formed between the subunits, mostly formed between side chains. The importance of interchain interactions is also reflected in the large relative interface and small relative buried surface areas.

Cluster 3 and 4 complexes exhibit similar features, including a balanced ratio of various secondary structure elements and polar/hydrophobic balance of various molecular surfaces and contacts. For both clusters, interchain contacts rely mostly on side chain–side chain and backbone–backbone contacts. The main difference between the two clusters is the relative role of the interface between the participating proteins. Cluster 3 members have a larger-than-average interface, in terms of both molecular surface and number of contacts, meanwhile cluster 4 complexes have a very restricted interface size, incorporating only a few atomic contacts.

Members of cluster 5 are the most similar to the average in most structural features. There are only weak distinguishing features, including a slightly increased helical content at the expense of extended structural elements, a moderate increase in the role of backbone–side chain interactions in interchain contacts, and the increased ratio of interchain contacts. However, these deviations in average parameter values are modest and—with the exception of the decreased extended structure content—none of them reaches 20% compared to the average values calculated for all complexes.

### 2.3. Defining Interaction Types Based on Sequence and Structure Clusters

Considering together the previously established sequence- and structure-based clusters, in total 20 types of complexes can be defined (Figure 3). The number of known complexes in possible types shows large variations, with some highly favored ones (e.g., type 2[sequence]/2[structure]) and ones with a single known example (e.g., type 2/1), showing that not all sequence compositions are compatible with all types of adopted structures. In order to arrive at a reasonable number of basic complex types, types with 10 or fewer complexes were either merged with the adjacent sequence clusters or were omitted. As structural differences in general are larger between clusters, types corresponding to different structure clusters were never merged. For structure clusters 1 and 2, only two adjacent sequence clusters were merged, as these contain over 95% and 85% of the complexes, respectively. In contrast, for structure classes 3 and 4, all four sequence clusters were merged, as the distribution of complexes is more even across the sequence space. For structure cluster 5, even a single sequence cluster is enough to capture over 85% of complexes, and thus no merging was employed. This approach yielded five main interaction types, each of which has over 20 complexes. In order to include all known MSF complexes, a sixth pseudo-type was introduced, which contains all structures not compatible with any of the previously described five types (see Appendix A for an exhaustive list).

The complex types defined so far are based on structure and sequence features. However, if these types represent biologically meaningful classes, there should be other relevant differences between them in terms of the energetics of the interaction, binding strength, subcellular localization, or the biological regulation of the interaction. In the next chapters, we describe each complex type with biologically important characteristics and assess the potential differences between the members of each class.

### 2.4. Complex Types Show Characteristic Energetic Properties

From a biological perspective, the strength of association between interacting protein chains and the stability of the resulting complex is of utmost importance. Unfortunately, complexes formed exclusively by IDPs via MSF generally lack targeted measurements concerning thermodynamic and stability parameters. However, low-resolution energy calculations and prediction algorithms can give an indication about the characteristic energetics properties of the uncovered complex types in general. While these methods might have fairly large errors in individual cases, they are well equipped for comparative studies between groups of complexes.

In order to assess the energetic properties of complexes, we employed an energy calculation scheme using low-resolution force fields based on statistical potentials (see Data and Methods). As a reference, energetic properties were calculated for complexes formed exclusively by ordered proteins and complexes formed by an IDP binding to an ordered partner via coupled folding and binding (CFB) (see Data and Methods and Appendix A). Figure 4 shows two types of calculated energies for each complex. On one hand, we calculated the total energy per residue in the whole complex, which reflects the overall stability. On the other hand, we also calculated the fraction of this stabilizing energy coming from intermolecular interactions (i.e., how important the interaction is for stability). In accordance with our expectations, complexes formed by ordered proteins feature strongly bound overall structures, with fairly large negative stabilizing energy/residue. In contrast, CFB complexes in general have less favorable per-residue energies, hinting at their comparatively weakly bound overall structures. However, the energetic feature providing the most recognizable difference between ordered and CFB complexes is the energy contribution of interchain contacts to the overall stability. In the case of ordered complexes, this contribution is fairly limited, as individual subunits have a stable structure on their own. In contrast, if the complex features an IDP, the interaction energy becomes a major contributor to stability (Figure 4a).

While ordered and CFB complexes tend to segregate in this energy space, complexes formed by MSF seem to be more heterogeneous, covering the whole available range of energetic values (Figure 4b). In the case of near-ordered proteins (Type 1), the energies resemble that of ordered complexes, hinting at the borderline ordered nature of the constituent IDPs, with the interaction between subunits playing a minor role. In contrast, coiled-coil-like structures (Type 2) on average have a much less stable complex structure, with interaction playing a substantial role in stability. These complexes resemble IDPs bound to ordered domains, and are expected to include several transient interactions. Other types fall largely between these two extreme cases. Energetics properties of the two types of oligomerization modules (Types 3 and 4) reflect the differences in interface surface area and contact numbers, shown in Figure 2. While the overall stability for both types varies in a very wide range, on average, the contribution of the interaction is higher for interface-heavy complexes (Type 3) than for interface-light ones (Type 4). Handshake-like folds (Type 5) show interesting properties: these complexes are quite stable with only limited variation in the per-residue energies. Yet, they achieve this high stability by relying heavily on the interaction between subunits of the dimer. As opposed to the complexes in Figure 4a, MSF complexes show high overlap in the energy space. This shows that very different structures, with potentially very different sequence compositions, can have similar energetic properties. Also, the high variability of energetic properties within complex types (the main reason for high overlap between different groups) shows that depending on the biological function, similar complexes can be required to have very different stabilities. For example, while several dimeric transcription factors can have similar structures that accommodate DNA-binding, the association and dissociation rates of the dimers (regulating their transcriptional activity) have to adapt to the required expression profiles of the genes they regulate.

The transient or obligate nature of interactions provides clues about their roles in biological systems. This is at least partially describable through K_d_ dissociation constants. While there is ample data about K_d_ values of IDPs binding via CFB to ordered domains [23], these values are largely missing for MSF complexes. In accord, we calculated estimated K_d_ values for MSF complexes (Appendix A), with Figure 5 showing the K_d_ distributions for the six previously defined complex types. In a biological context, actual K_d_ values can be a nonlinear function of environmental parameters. Unfortunately, this information is largely unknown for most MSF complexes, and such predicted K_d_ values should be treated with caution and should only be used for comparing group averages, where individual errors can even out. The lowest average K_d_ values were calculated for complexes with a handshake-like fold (Type 5). The next two types with low K_d_s are the near-ordered complexes (Type 1) and interface-heavy oligomerization modules (Type 3). These three types together possibly cover most cases of the interactions where the complex needs to stay stable for an extended period of time, such as histone dimes (Type 5), complexes with enzymatic activity (Type 1) and several transcription factors (Type 3). Coiled-coil-like structures and oligomerization modules with small interfaces in general have a higher K_d_, indicating that several transiently bound complexes belong to these types.

### 2.5. Interactions Are Heavily Regulated by Several Mechanisms

While the energetics of various interactions can provide clues about their transient/obligatory nature, the regulatory mechanisms can give more direct evidence. For example, while most IDP enzymes (belonging to Type 1) form particularly stable oligomers, indicating an obligate interaction, for example the oligomeric state of superoxide dismutase (SOD1) is known to be controlled by post-translational modification (PTM) serving as an on/off switch [35]; meaning that despite a strong interaction, it is reversible, and the disordered state of the monomers is biologically relevant (Figure 6a). Figure 6a shows additional examples of various regulatory mechanisms of MSF interactions via PTMs. These regulatory steps have already been described in the case of IDPs that bind to ordered domains [36], but have not been studied in the context of IDPs participating in MSF interactions. Apart from the on/off switch exemplified by SOD1, PTMs can control the partner selection of synergistically folding IDPs, such as in the case of another tightly bound complex, formed by H3/H4 histones (Type 5) [37]. PTMs can also tune the affinity of certain interactions, as is the case for the activating p53/CBP interaction (Type 4) [38]. Apart from these mechanisms that directly control the interaction between IDPs, PTMs can have a more indirect effect, modulating the activity of the dimer itself. In the case of the Max dimeric transcription factor, phosphorylation at the N-terminus of the binding region controls the dimer’s (Type 4) interaction capacity towards DNA [39]. An even more indirect modulation of function is displayed for the retinoblastoma protein Rb, which in complex with E2F1/DP1 (Type 3) has a strong transcriptional repression activity. Upon methylation, Rb recruits L3MBTL1 [40], which is a direct repressor of transcription via chromatin compaction, augmenting the effect of Rb through a related but separate mechanism extrinsic to the Rb/E2F1/DP1 complex. This way the strength of repression depends on the PTM of the MSF complex, but through an additional protein that is not part of the complex but contributes to the complex function through a parallel mechanism in an indirect way.

To have a more systematic picture of the extent of regulatory mechanisms in MSF interactions, Figure 6b shows the fraction of known MSF complexes with experimentally verified PTM sites (Appendix A). In total, nearly 30% of studied complexes feature at least one PTM that was experimentally verified in a low-throughput experiment, presenting a regulatory mechanism that is able to directly or indirectly modulate either the interaction itself, or the activity of the resulting complex. The most prevalent PTM is phosphorylation, affecting 22% of complexes, but 10%, 15%, and 5% of MSF complexes contain methylation, acetylation, and ubiquitination sites as well (Figure 6b).

In addition, complex formation can also be regulated through the availability of the subunits participating in the interaction. This availability can depend on the alternative mRNA splicing of the corresponding genes, where certain isoforms lack the binding site (Appendix A). Also, even if the translated isoform has the binding site, the protein itself can be sequestered by competing interactions with other protein partners (Appendix A). These mechanisms are present for 11% (alternative splicing) and 16% (competing interactions) of complexes, and together with PTMs, in total 36% of MSF complexes have at least one known regulatory mechanism for modulating the interaction. Furthermore, these regulatory mechanisms often act in cooperation, with seven interactions known to employ PTMs, alternative splicing, and competing interactions as well (Figure 6c).

### 2.6. Various Complex Types Show Differential Subcellular Localization

In addition to regulatory mechanisms detailed in the previous chapter, a crucial element in the spatio-temporal control of protein function is subcellular localization [41]. In order to assess this aspect of MSF complexes, and to understand if the defined interaction types have different properties in terms of cellular localization, we used “cellular component” terms from GeneOntology (GO) [42] (see Data and Methods). Various GO terms were condensed into five categories including “Extracellular”, “Intracellular”, “Membrane”, “Nucleus”, and “Other” to enable an overview of the differences in localization between the six complex types (Figure 7) (for exact GO terms for each complex see Appendix A).

The least amount of information is available for Type 1, near-ordered complexes. Albeit GO terms are lacking for most complexes, even the limited annotations highlight that these complexes are able to efficiently function in the extracellular space, which in general is fairly uncommon for IDPs. Coil- and zipper-type helical complexes (Type 2) are somewhat more often attached to the membrane or function in the intracellular space, or in non-nuclear environments, such as the lysosome. In contrast, oligomerization modules (Types 3 and 4) are most prevalent in the nucleus and the intracellular space, which is in line with the function of the high number of transcription factors in these groups. However, modules with a large interface (Type 3) are relatively often found in other compartments, while modules with smaller interfaces (Type 4) also function in the extracellular space. Complexes adopting a handshake-like fold are enriched in histones, which is reflected in their enrichment in the nucleus and the chromatin (classified as “other” in Figure 7). Type 6 complexes are heterogeneous in terms of localization as well, and hence members can be found in all studied localizations to a comparable degree. These preferences in subcellular localization for different complex types reinforce our notion that even though our classification scheme relies on sequence and structure properties alone, the obtained interaction types also have biological meaning.

### 2.7. The Annotated Catalogue of Complexes Formed via Mutual Synergistic Folding

Considering the previously analyzed features of complexes, averaging the calculated features for the six established interaction types provides the annotated catalogue of MSF interactions (Figure 8). Apart from the main sequential and structural features, Figure 8 also shows example structures, energetic properties, subcellular localization, and the main regulatory mechanisms for each complex type.

The first type of complexes bears a high similarity to ordered protein complexes, and hence are named near ordered. The constituent chains are usually similar, in many cases corresponding to homooligomers, with a high Pro/Gly content and typically only a few charges. The main difference compared to protein complexes formed by ordered proteins is that near ordered subunits are depleted in α-helices [28]. For reaching a stable structure through the interaction, they utilize a large number of intrachain contacts, with inter-subunit interactions through a small polar interface playing only a secondary role in the stability of the complex. This group contains a large number of enzymes, transport proteins, and nerve growth factors, where the exact structure is of utmost importance; however, in contrast to monomeric proteins, the presence of this structure relies on the interaction. This interaction type is mostly regulated through phosphorylation and acetylation of binding site residues. These proteins resemble ordered proteins in their localization as well, with extracellular regions being highly representative.

The second type of complexes contains structures with a high overall similarity, mostly consisting of coiled-coils and zippers, structures composed of parallel interacting helical structures, often stabilized by a restricted set of residues, such as leucines, alanines, or tryptophans. In general, constituent proteins are depleted in residues incompatible with α-helix formation, such as Pro and Gly, and also in aromatic residues. In turn, they are abundant in hydrophobic residues and show an enrichment for either polar or charged residues. The constituent helices usually form a fairly weakly bound system, where the interchain interactions via the relatively large interfaces play a major role. Constituent proteins are able to bury only a small fraction of their polar surfaces. Coiled-coil interactions are often regulated, typically via various types of PTMs, most often through phosphorylation or, to a lesser degree, acetylation. Despite their highly similar structures, complexes in this group convey a large variety of functions, mainly pertaining to regulating transcription and performing membrane-associated biological roles, such as organelle and membrane organization.

The third and the fourth type of complexes are both generic oligomerization modules that can be split according to the importance for the interchain interactions, grouping them as either interface-heavy (Type 3) or interface-light (Type 4) complexes. In both cases, the sequences can be highly variable, and the unifying features are mostly structural. Both types typically have an average-sized relative buried area with balanced hydrophobic/polar composition. However, interface-heavy complexes have a large, slightly polar interface that plays a major role in achieving the tightly bound structures. In contrast, interface-light complexes form a more helical structure and have smaller hydrophobic interfaces that play a more diminished role in achieving the stability of a less tightly bound system. This hints at interface-light complexes being more transient, also supported by the fact that these complexes have a higher number of known regulatory PTMs and are also modulated by alternative splicing. Both type 3 and type 4 complexes preferentially occur in nuclear and intracellular processes, as several of them are ribbon–helix–helix (interface-heavy) or basic helix–loop–helix (interface-light) transcription factors, able to shuttle between the nuclear and the intracellular spaces. In addition to the similarities in subcellular localization, type 4 complexes preferentially occur in the extracellular space, and type 3 complexes in other cell compartments, as well.

The fifth type of complexes typically adopts a handshake-like fold, characteristic of histones and homologous proteins. While these structures are usually largely helical, the interacting proteins often contain a relatively high ratio of prolines and glycines, in addition to the enrichment of aromatic residues. While they are depleted in polar residues, both the interface and the buried surface have a fairly balanced hydrophobic/polar makeup. The complexes are relatively tightly bound, and interchain interactions play a fairly large role in stabilizing the interaction. This type of complex has the highest ratio of both PTMs and competitive interactions, providing a large amount of regulation. In addition, PTMs are highly heterogeneous, containing phosphorylations, acetylations, methylations, and ubiquitinations as well. Members of this cluster primarily serve DNA/chromosome-related functions, and hence are usually located in the nucleus.

While types 1–5 represent well-defined groups with members of clear unifying similarities, the final group serves as an umbrella term for complexes that are not members of any previous structural/sequential class. In accord, these complexes cannot be described by simple characteristic features and are the most sequentially and structurally heterogeneous group. This group contains highly specialized interactions that present unique protein complexes, which are regulated through all three control mechanisms and occur in all studied subcellular localizations.

### 2.8. Interaction Types Present A Novel Classification of Protein Complexes

The described MSF classification method bears similarity to the approach employed in CATH, as both approaches use a hierarchical classification of PDB structures. However, CATH does not consider interactions and simply relies on the secondary structure elements and their connectivity and arrangement, in contrast to the presented analysis taking into account protein chain interactions too, together with sequence composition features.

Figure 9 shows the studied MSF complexes in both our MSF classification system and in CATH, considering the top two levels (“Class” and “Architecture”). The highest-level CATH definitions, corresponding to “Class”, reflect the overall secondary structure element distribution of the structures. In this framework, Type 1 near-ordered complexes mostly occupy the “Mainly Beta” CATH class, while complexes from the other five types mostly fall into the “Mainly Alpha” class or the “Other” class. At the next CATH level, “Architecture”, certain MSF type complexes (such as type 2 coils and zippers) are segregated into further subclasses.

Considering “Class” and “Architecture” definitions, there is very little correspondence between the CATH and the new MSF classification. If the two schemes showed a high degree of similarity, the matrix in Figure 9 should be close to a diagonal matrix. In reality, however, off-diagonal elements are large, confirming the novelty of the presented MSF classification scheme.

## 3. Discussion

Here, we present the first approach aiming at the classification of complex structures formed exclusively by disordered proteins via mutual synergistic folding. We developed and applied a method that can classify these complexes into various types based on sequence- and structure-based properties. The classification scheme takes into account on the one hand, the overall sequence and structure properties of the complex, and on the other hand, the interaction itself, quantifying the role of intra- and intermolecular interactions in relation to the overall contact/surface properties of the structure. As the classification protocol is based on hierarchical clustering, it is freely scalable. Tuning the resolution via changing the number of sequence-based or structure-based clusters, the method can be used to yield any number of types and subtypes. The presented classification is a top-level one highlighting the major types of MSF classes, and this six-way classification scheme will be used to better define MSF complex types in the MFIB [26] database.

While both sequence- and structure-based parameters are taken into account when defining the final complex types, the two sets of descriptors have different roles in the scalability of the method. In our presented approach to defining complex types, the main features are structural properties, while sequence parameters are more descriptive in the sense that they highlight the sequential features needed to be able to fold into a complex of given structural properties (Figure 3). However, sequence features can be used to distinguish subtypes of structure-defined complex types. For example, type 1 near-ordered complexes come in two flavors according to the two sequence clusters they cover (Figure 1 and Figure 3): polar-driven interactions between mostly homodimers, and charge/hydrophobic driven interactions between mostly heterodimers. Also, type 2 complexes (coils and zippers) come in two varieties: relying on polar-driven interactions for heterodimers and charge-driven for homodimers.

In addition to providing a scalable classification scheme, the described method and the defined complex types have biological relevance. The presented complex types have different biological properties; although only information describing the sequence and structure properties were put in, the resulting types show different properties in terms of the energetics and strength of the interactions (Figure 4 and Figure 5), the relevant regulatory processes (Figure 6), and subcellular localization (Figure 7).

The analysis of the energetics properties of the interactions can provide a glimpse into the biophysical details of the binding and folding. The use of low-resolution statistical force fields proved to be a suitable approach to discriminate complexes based on the structural features of constituent chains [28] and to describe the binding of IDPs [43,44]. While complexes of ordered proteins and domain-recognition IDP binding sites have a fairly narrow range in energetics parameters (Figure 4a), complexes formed exclusively by IDPs are more heterogeneous, basically covering the whole range of the energy spectrum (Figure 4b). Furthermore, based on predictions, MSF complexes cover at least 10 orders of magnitude in K_d_ values (Figure 5). Hence, in terms of binding strength and stability, these complexes have the potential to cover a very wide range of biological functions, overlapping with those of ordered complexes and domain-binding IDPs as well, in agreement with the previous comparative functional analysis of a wide range of interactions [28].

For most known MSF complexes, the resulting structure is instrumental for proper function, such as the coiled-coil structure for the SNAP receptor (SNARE) complex in mediating membrane fusion [45], the dimeric structure for a wide range of transcription factors in precise DNA-binding [46,47,48], and the proper coordination of catalytic residues for oligomeric enzymes [49,50]. Therefore, for MSF complexes, the interaction de facto switches on the protein function, and hence the precise regulation of the interaction strength is vital in the biological context of these complexes. While structure-based K_d_ value predictions are informative, in some cases they do not fully describe the interactions. Many MSF complexes are tightly bound, yet they are not necessarily obligate complexes, and their association/dissociation can be under heavy regulation. For example, solely based on K_d_ values and energetics, type 5 (handshake-like fold) interactions seem to form obligate complexes. However, there are several cases where these interactions do break up in a biological setting, most notably for histones. Histone H4 is able to form dimers with at least eight different H3 variants [51], and it was described that in the case of H3.1 and H3.3, the preference of H4 for these two partners is governed by H4 phosphorylation [37]. The post-translational modifications can enhance complex formation or dissociation in many other cases as well [35]. In addition, competition for the same binding partner and binding site availability as a function of alternative splicing is an additional mechanism for the regulation of the formation of MSF complexes (Figure 6).

Exploring the precise regulatory mechanisms for MSF complexes would be highly informative. Unfortunately, experimental K_d_ measurements are lacking for the majority of these interactions, and interactions in structural detail have usually been only analyzed in a single PTM state. Therefore, the molecular details and biologically relevant steps of the regulation of these interactions are difficult to assess; but from a biological sense, it is probable that even several low K_d_ complexes can dissociate rapidly in certain cases. At least some regulatory mechanisms are currently known for about 36% of studied MSF complexes, but the real numbers are bound to be higher. This means that most probably the majority of MSF complexes are not obligate complexes, where the disordered state is physiologically irrelevant, but can exist in both the stable bound state and the disordered unbound state as well, under native conditions. Thus, MSF complexes are integral parts or direct targets of regulatory networks, although the extent of regulation varies with the interaction type considered.

Apart from the studied regulatory mechanisms, additional layers of spatio-temporal regulation can play crucial roles for MSF complexes, similarly to other IDP interactions [41]. An emerging such regulatory mechanism is liquid–liquid phase separation (LLPS). A prime example is the Nck/neuronal Wiskott–Aldrich syndrome protein (N-WASP). N-WASP is known to undergo LLPS when interacting with Nck and nephrin [52], via linear motif-mediated coupled folding and binding. Mutually synergistic folding between the secreted EspFU pathogen protein from enterohaemorrhagic *Escherichia coli* and the autoinhibitory GTPase-binding domain (GBD) in host WASP proteins (MFIB ID:MF2202002, type 5 complex) hijacks the native LLPS-mediated cellular processes [53], showing that competing interactions are not always stoichiometric in nature, and the true extent of MSF regulation is likely to be even more complex than highlighted here.

The difference between complex types in various biological and biophysical properties shows that these type-definitions reflect true biological differences. Apart from being useful for complex classification, the presented method also shows that differences in binding strength, subcellular localization, and regulation are encoded in the sequence and structural properties of proteins. This can be the basis for developing future prediction methods, where these sequence- and structure-based parameters can be used as input for the prediction of biological features of complexes. In addition, the establishment of MSF complex types has direct implications, as knowledge present for a specific complex might be transferable to other complexes of the same type. For example, certain pathological conditions arise through the aggregation of IDPs. A well-known example is transthyretin (TTR) aggregation that can lead to various amyloid diseases, such as senile systemic amyloidosis [54]. Another example from the same near-ordered complex type is the superoxide dismutase SOD1, which is able to form aggregates in amyotrophic lateral sclerosis [55]. While the localization and the biological function of TTR and SOD1 (hormone transport and enzymatic catalysis) are radically different, their potency of malfunctioning (often connected to various mutations) share a high degree of resemblance. On one hand, this marks other type 1 complexes as candidates for toxic aggregation, on the other hand, it indicates that the potential therapeutic techniques for one complex (e.g., CLR01 for TTR) can give clues about potential targeting of other interactions.

Such structural classification approaches can have a high impact on structure research, most importantly in the study of protein structure or evolution, in training and/or benchmarking algorithms, augmenting existing datasets with annotations, and examining the classification of a specific protein or a small set of proteins [56]. Up to date, several structure-based classification approaches have been developed, such as SCOP [32] and CATH [33], which are extended to protein complexes as well. In this sense, previously existing methods are able to classify MSF complexes too. However, the approaches used do not take into account that these structures are only stable in the context of the interaction, and that a certain protein region can adopt fundamentally different structures depending on the interacting partner. The lack of the explicit encoding of parameters describing the properties and importance of the interaction into the classification scheme makes current methods unable to accurately describe the spectrum of MSF complexes, and to date, no such dedicated classification scheme has been proposed. In contrast to previously existing methods that largely encode the same information [57], the presented MSF classification scheme is highly independent (Figure 9), and thus serves as an orthogonal approach capable of properly handling the specific properties of IDP-driven complex formation through mutual synergistic folding.

## 4. Data and Methods

### 4.1. Complexes Formed Through Mutual Synergistic Folding (MSF)

MSF complexes were taken from the MFIB database [26]. Two entries, MF2100018 and MF5200001, from the 205 were discarded due to issues with the corresponding PDB structures 1ejp and 1vzj, as constituent chains have an unrealistically low number of interchain contacts. Problems with these two structures are apparent from the high outlier scores and clash scores provided on the PDB server. As the developed classification scheme relies heavily on structural parameters, we opted to leave these two entries out of the calculations. The final list of entries is given in Appendix A.

### 4.2. Other Complexes of Ordered and Disordered Proteins

As a reference, two other datasets of protein complexes were used. A set of complexes formed exclusively by ordered single-domain protein interactors was taken from [28]. These 688 complexes (see Appendix A) are formed via autonomous folding followed by binding, that is, both interacting protein chains adopt a stable structure in their monomeric forms, prior to the interaction. A set of 772 complexes with an IDP interacting with ordered domains was taken from the database of Disordered Binding Sites (DIBS) database [23]. These complexes (see Appendix A) are formed via coupled folding and binding, where the IDP adopts a stable structure in the context of the interaction. 

### 4.3. Calculating Sequence Features

Similarly to the approach described in [28], the following amino acid groups were used in quantifying sequence composition of proteins: hydrophobic (containing A, I, L, M, V), aromatic (containing F, W, Y), polar (containing N, Q, S, T), charged (containing H, K, R, D, E), rigid (containing only P), flexible (containing only G), and covalently interacting (containing only C). This low-resolution sequence composition at least partially compensates for commonly occurring amino acid substitutions that in most cases do not affect protein structure and function. In all cases, compositions were calculated for the entire complex, including all interacting protein chains. An 8th sequence parameter was used to quantify the compositional difference between subunits. This dissimilarity measure was defined as:

$\Delta_{total}=\sum_{i=1}^{7}\Delta_{i}$, where $\Delta_{i}\;$ is the largest composition difference of residue group *i* between any pair of constituent chains. The average dissimilarities for various sequence-based clusters are shown in Figure 1. For exact sequence composition values for all MSF entries, see Appendix A.

### 4.4. Calculating Structure Features

Secondary structure assignment was performed by DSSP [58], using a three-state classification distinguishing helical (’H’,’G’,’I’), extended (’B’,’E’), and irregular (’S’,’T’, unassigned) residues.

Molecular surfaces were calculated using Naccess [59]. Solvent accessible surface area (SASA) was defined by the Nacces absolute surface column. Interface is defined as the increase in SASA as a result of removing interaction partners from the structure. Buried surface was calculated by subtracting interface area and SASA from the sum of standard surfaces of residues in the protein chain. Thus, interface and buried surfaces represent the area that is made inaccessible to the solvent by the partner(s) or by the analyzed protein itself. All calculated areas were split into hydrophobic (H) and polar (P) contributions based on the polarity of the corresponding atom. Polar/hydrophobic assignations were taken from Naccess.

Contacts were defined at the atomic level. Two atoms were considered to be in contact if their distance is shorter than the sum of the two atoms’ van der Waals radii plus 1 Angstrom. For exact structural feature values for all MSF entries, see Appendix A.

### 4.5. Filtering Features for Clustering

Standard Pearson correlation values were calculated between all sequence and structure features (Appendix A). If two features show a correlation with an absolute value above 0.7, only one was kept. In each case, we discarded the feature that shows a high correlation with a higher number of other features, or the one with the lower standard deviation. In total, none of the seven sequence parameters were discarded, but 13 out of the 24 structure parameters were omitted from subsequent clustering steps.

### 4.6. Clustering

Both sequence and filtered structure parameters were used as input for clustering separately. First, hierarchical clustering was done using the scaled features as input, using Euclidean distance and Ward’s method (Appendix A). Then, k-means clustering was employed, and the within-groups sum of squares were plotted as a function of the number of clusters (Appendix A). k-means clustering analysis did not provide a clear-cut support for the number of clusters to choose, and hence we opted for choosing a low number of clusters in both cases (four and five in the case of sequence- and structure-based clustering, respectively), that are not in contradiction with the k-means analysis. This choice of cluster numbers reflects our preference for providing an overall high-level classification. Clustering was done using R with the Ward.D2 and k-means packages.

### 4.7. Energetic Features

Interaction energies for residues were calculated using the statistical potentials described in [60]. These interaction potentials were demonstrated to well describe the energetic features of IDP interactions [43], and are the basis for recognizing them from the sequence [44]. These potentials yield dimensionless quantities in arbitrary units, and hence their absolute values bear no direct physical meaning. However, their signs are accurate, and values below 0 correspond to stabilizing interactions. Furthermore, they can be directly compared, and hence more negative values typically correspond to more stable structures. In each analysis, the total energies were calculated from the residue-level interactions from the entire complex. Two residues were considered to be in interaction if there is at least one heavy atom contact between them. Energetic values are given in Appendix A.

### 4.8. Prediction of K_d_ Values

Dissociation constants for MSF complexes were estimated using the method described in [61]. In each case, the modified PDB structures taken from the MFIB database [26] were used as input. For technical reasons, not all structures yield a K_d_ value prediction, and thus the number of values used in representing the average per-complex type K_d_s (Figure 5) is calculated from fewer values than the actual number of complexes per type. K_d_ values are listed in Appendix A.

### 4.9. Post-Translational Modifications (PTMs), Isoforms and Competitive Binding

Post-translational modifications were taken from the 2 October 2017 version of PhosphoSitePlus [62], PhosphoELM [63], and UniProt [64]. Only PTMs that were identified in low-throughput experiments were used. These were mapped to complex structures using BLAST between UniProt and PDB sequences (Appendix A). Protein isoforms were taken from the 4 October 2017 version of UniProt (Appendix A). To determine alternative binding partners for IDPs, all oligomer PDB structures containing the same UniProt region were selected. PDB structures listed as related in the corresponding MFIB entry were removed. Structures containing the same interaction partners as the original complex were also removed (Appendix A).

### 4.10. GeneOntology Terms for Assessing Subcellular Localization

Subcellular localization was represented using GeneOntology [42] terms from the cellular_component namespace. Terms attached to complexes in MFIB were mapped to a restricted set of terms, called CellLoc GO Slim, used in previous studies [28] to compare localization of protein–protein interactions. Terms in CellLoc GO Slim were split into five categories: extracellular, intracellular, membrane, nucleus, and other, encompassing other membrane-bounded cellular compartments, such as the lysosome, as well as non-membrane-bounded compartments, such as the chromatin. For CellLoc GO terms attached to MSF complexes, see Appendix A.

## Figures and Tables

**Figure 1 ijms-20-05460-f001:**
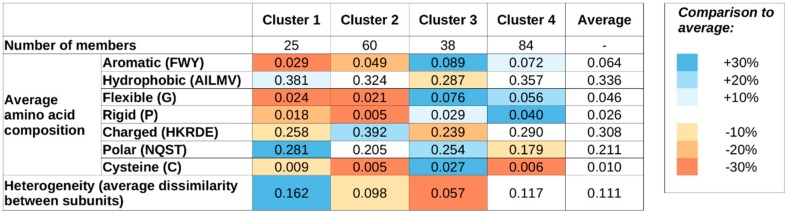
Average values of sequence features for the four sequence-based clusters. Blue and orange shadings mark values that are over- or under-represented compared with the average of all MSF complexes. Heterogeneity values were not used for cluster definitions.

**Figure 2 ijms-20-05460-f002:**
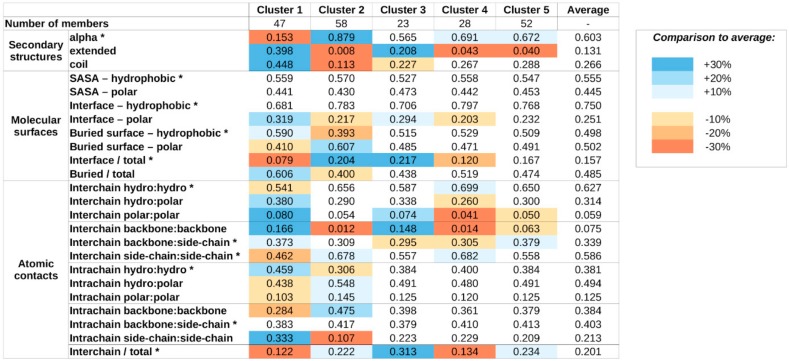
Average values for structure features for the five structure-based clusters. Blue and orange shadings mark values that are over- or under-represented compared to the average of all MSF complexes. SASA—solvent accessible surface area, hydro:hydro—fraction of contacts that are formed between two hydrophobic atoms. Asterisks mark features that were included in the clustering.

**Figure 3 ijms-20-05460-f003:**
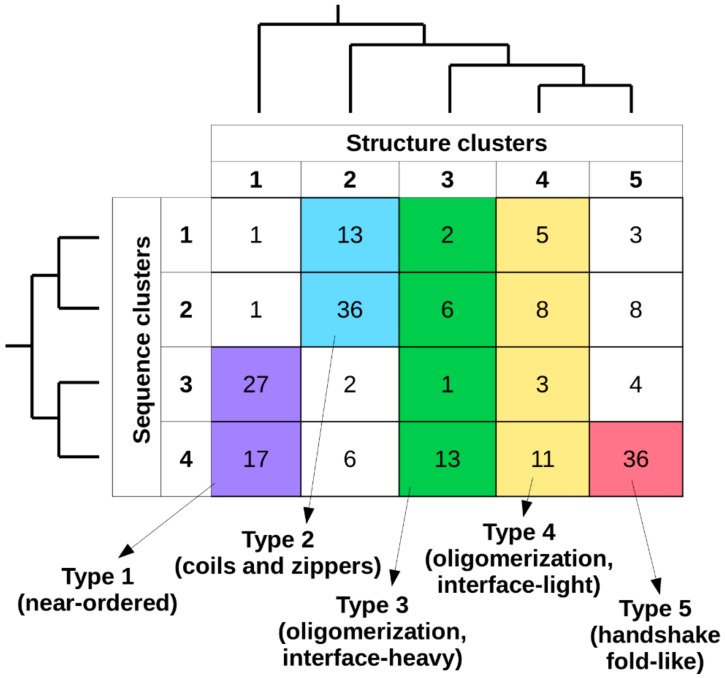
MSF complex types. Colored regions mark separate interaction types considering sequence- and structure-based clusters (vertical and horizontal axes, respectively). The relationship of each sequence-and structure-based cluster taken from the hierarchical clustering (Appendix A) is shown on the corresponding side of the table. Each of the six defined types is assigned a randomly selected color (that is of high contrast), and these are used in later figures to denote the corresponding complex types.

**Figure 4 ijms-20-05460-f004:**
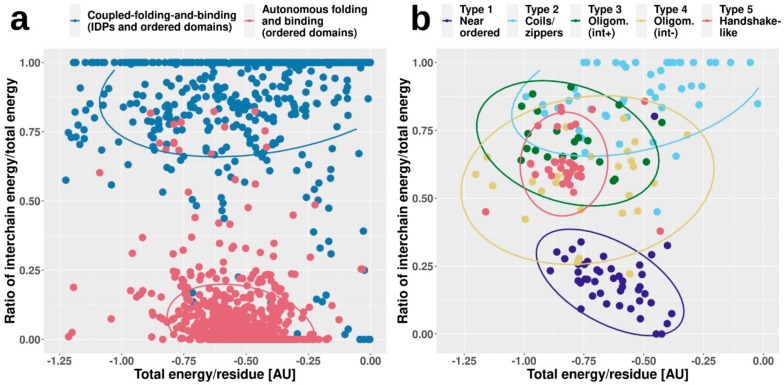
Energetic parameters of various interaction classes. The relative energetic weight of intersubunit interactions in the overall stability (*y*-axis) as a function of the overall energy per residue (*x*-axis, measured in arbitrary units, AU) for ordered complexes and complexes formed by coupled folding and binding (**a**), and the five well-defined types of MSF complexes (**b**).

**Figure 5 ijms-20-05460-f005:**
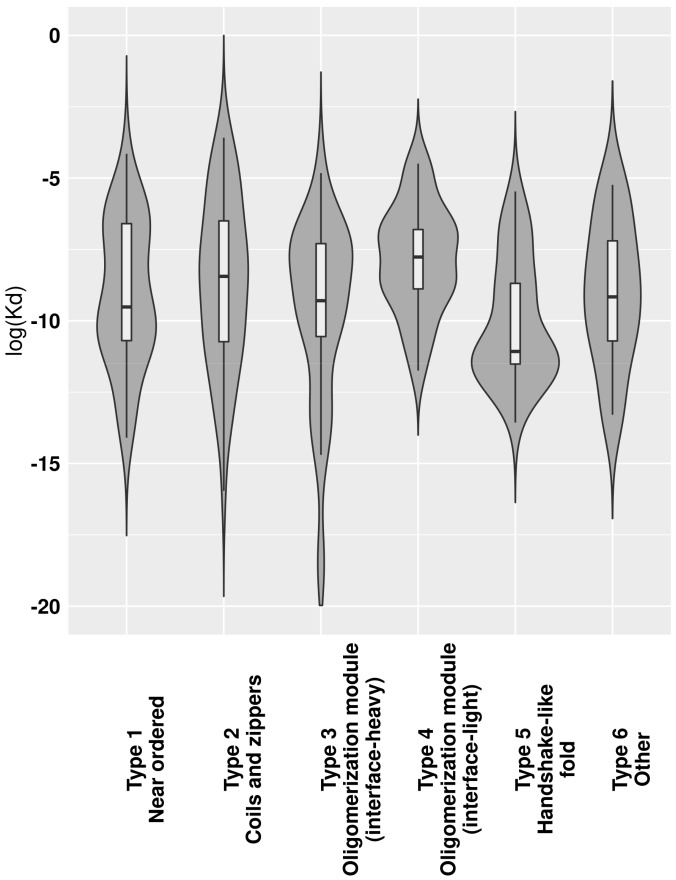
Predicted K_d_ value distributions for the six types of MSF complexes.

**Figure 6 ijms-20-05460-f006:**
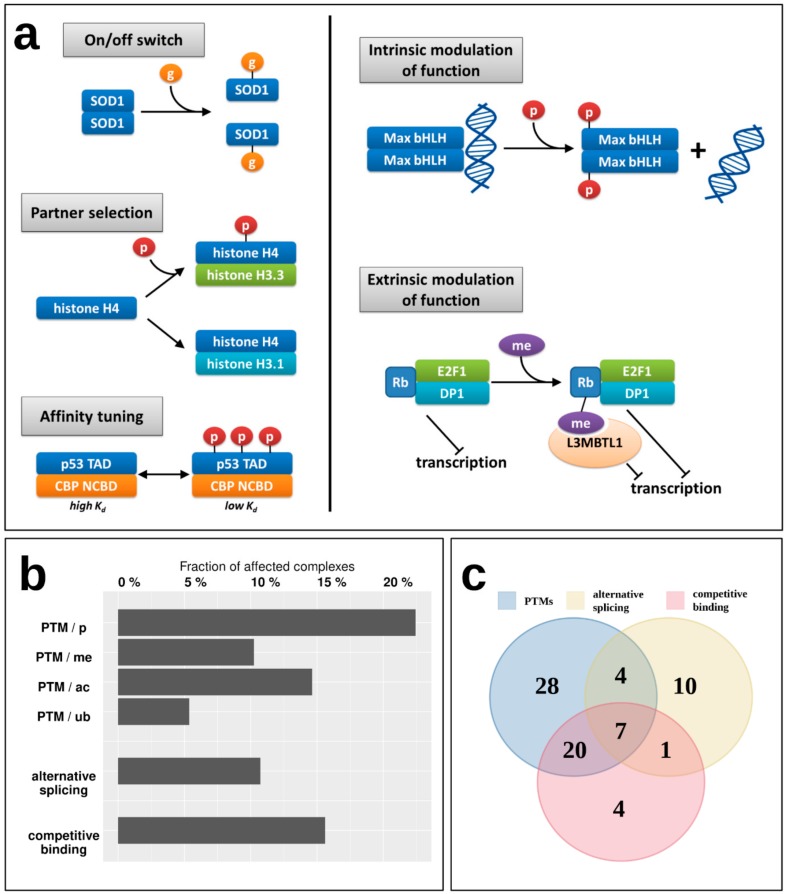
Regulatory mechanisms of MSF complexes. (**a**) examples of regulation and modulation of function through post-translational modifications. p—phosphorylation, g—glutathionylation, me—methylation, SOD1—superoxide dismutase, CBP—CREB-binding protein, Rb—retinoblastoma-associated protein. Colored boxes represent interacting chains forming the MSF complexes. (**b**) The fraction of complexes with verified PTM sites, and the fraction of complexes where at least one interactor is regulated via alternative splicing or by competing interactions. (**c**) Number and overlap of MSF complexes affected by the three types of regulatory mechanisms.

**Figure 7 ijms-20-05460-f007:**
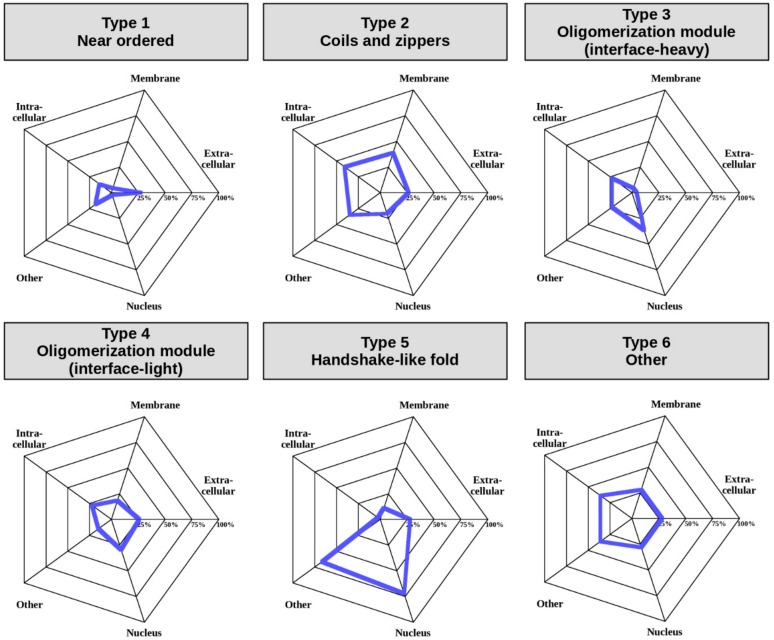
Subcellular localization of MSF complexes belonging to the six types. “Other” contains the “non-membrane-bounded organelle”, “secretory granule”, “lysosome”, “cytoplasmic vesicle lumen”, and “transport vesicle” GeneOntology terms.

**Figure 8 ijms-20-05460-f008:**
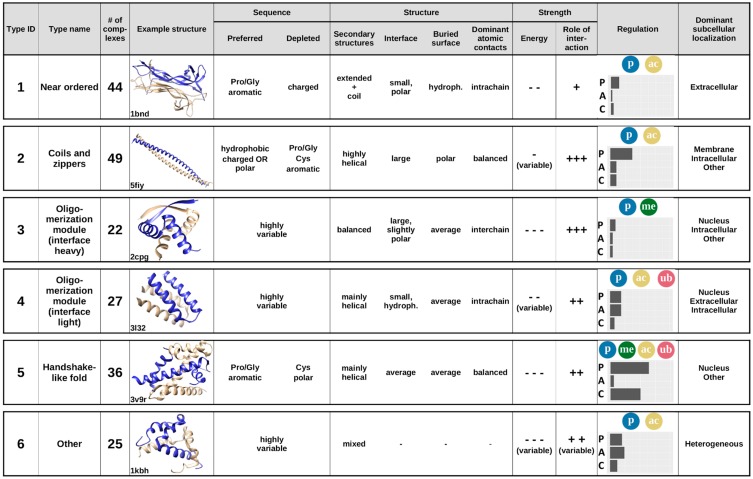
Annotated types of complexes formed by IDPs, based on sequence and structure features. Horizontal bars in the regulation column show the fraction of complexes in a given group involved in various types of regulatory mechanisms (P—post-translational modifications, A—alternative splicing affecting binding regions, C—competing interactions). Color circles mark the dominant post-translational modification(s) for the group (p—phosphorylation, me—methylation, ac—acetylation, ub—ubiquitination).

**Figure 9 ijms-20-05460-f009:**
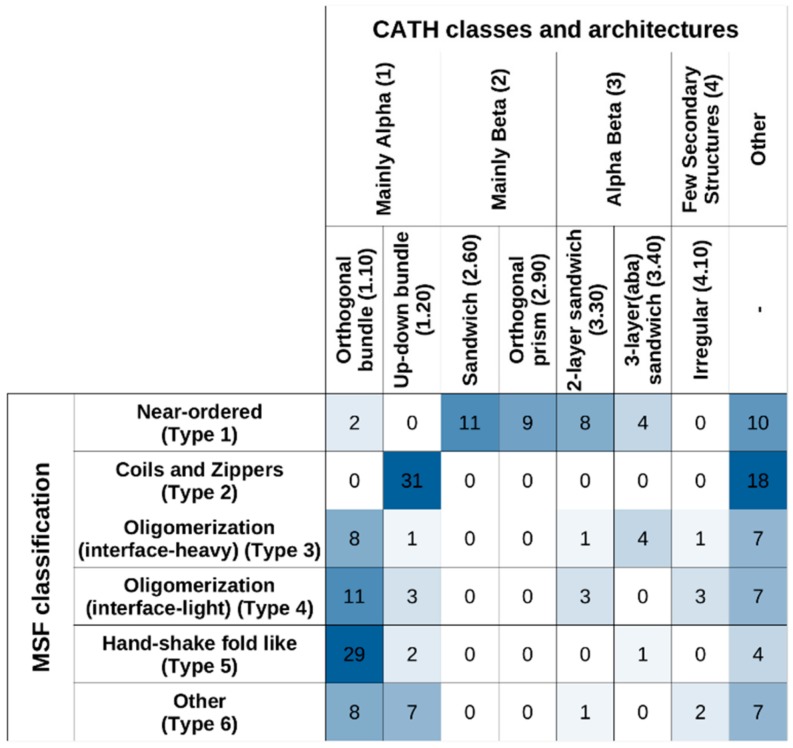
Overlap between CATH and MSF classification.

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
