# Peer review of "Sequence and Structure Properties Uncover the Natural Classification of Protein Complexes Formed by Intrinsically Disordered Proteins via Mutual Synergistic Folding"

_ijms, 2019, doi:10.3390/ijms20215460_

Round 1

Reviewer 1 Report

A nicely done article that introduces a novel classification scheme for protein complexes formed by IDPs. The paper is well-written and provides a well-rounded and compelling analysis of several different types of complexes that were formulated by the authors. I am looking forward to seeing this article published. I include a few comments which may help with strengthening presentation of this work.

The literature review in the first paragraph in the introduction, especially concerning the aspects related to protein-protein interactions, seems to rely on older literature, while a number of more recent and arguably more adequate results were published since. The clustering study requires a more careful justification and/or adjustments. The selection of the 4 clusters (Figure S2) is questionable – from my experience this plot shows that 9 clusters is a more appropriate choice. The figure does not provide a compelling justification for the selection of 4 and the authors did not provide an alternative rationale in text (section 4.6). My reservations are somehow supported by the fact that clusters 1 and 2 seems rather similar (Table 2) and perhaps should be further subdivided. Similarly, I disagree with the characterization that Figure S4 does not suggest a suitable number of clusters. Two most suitable choices are 6 or 11 (leaning towards the latter). Consequently, clusters 3 and 4 (Table 2) seems rather similar. These choices have a fundamental effect on the remainder of the article and should be better motivated or chosen. The bottom line is that the authors prefer a small number of clusters to secure a reasonably low number of complex types in Figure 1. While this is a reasonable rationale, the authors focus their justification solely on the quality of the underlying clustering which (in my view) does not justify the selection of such low number of clusters. I am assuming that at some point (after publication) the resulting classification will be annotated in the MFIB database, to facilitate future comparative studies.

Author Response

We would like to thank the reviewer for the careful reading of the manuscript and the provided positive and constructive comments. The manuscript was revised in light of these comments, and we believe that these corrections have improved the manuscript, making the main messages of our work more accessible. After each of the reviewer’s comments we address it in detail (marked in italics).

The literature review in the first paragraph in the introduction, especially concerning the aspects related to protein-protein interactions, seems to rely on older literature, while a number of more recent and arguably more adequate results were published since.

We revised the text and added more recent relevant publications, including papers on IDP functionality, interaction capacity and analysis of MSF interactions.

The clustering study requires a more careful justification and/or adjustments. The selection of the 4 clusters (Figure S2) is questionable – from my experience this plot shows that 9 clusters is a more appropriate choice. The figure does not provide a compelling justification for the selection of 4 and the authors did not provide an alternative rationale in text (section 4.6). My reservations are somehow supported by the fact that clusters 1 and 2 seems rather similar (Table 2) and perhaps should be further subdivided. Similarly, I disagree with the characterization that Figure S4 does not suggest a suitable number of clusters. Two most suitable choices are 6 or 11 (leaning towards the latter). Consequently, clusters 3 and 4 (Table 2) seems rather similar. These choices have a fundamental effect on the remainder of the article and should be better motivated or chosen. The bottom line is that the authors prefer a small number of clusters to secure a reasonably low number of complex types in Figure 1. While this is a reasonable rationale, the authors focus their justification solely on the quality of the underlying clustering which (in my view) does not justify the selection of such low number of clusters.

We agree that there is very little objective support for the exact choices of the numbers of clusters (4 for sequence and 5 for structure). Apart from a few rare cases, in biology there is seldom one absolutely correct solution for these clustering problems. We agree that from a clustering point of view larger cluster numbers are perfectly rational choices. However, choosing 9 and 11 clusters would yield in total 99 possible complex types, which would mean that we have about 2 clusters per type on average. Instead, we opted to have a more restricted number of final types (4*5=20 types, with 10 complexes on average - enough to calculate group averages).

We fully agree that this is more of a conceptual choice, and that we should make it clearer in the manuscript that this reflects our personal preference to achieve a high level classification scheme with restricted numbers of complex types. In accord, we added text to sections 2.1, 2.2 and 4.6.

I am assuming that at some point (after publication) the resulting classification will be annotated in the MFIB database, to facilitate future comparative studies.

Yes, we will adopt the new classification scheme into MFIB annotations instead of the rather ad hoc one currently used. A sentence has been added to the Discussion section to reflect this.

Reviewer 2 Report

In this well-written manuscript, Simon et al. presents a thorough study wherein a clustering-based classification scheme based on amino acid sequences and structural properties was developed to group complexes of intrinsically disordered proteins (IDPs) formed via mutual synergistic folding (MSF). Using this method, six major types of MSF complexes with corresponding biological significance were identified. Furthermore, this method also demonstrated that the sequence and structural properties of proteins could predict differences in binding affinities, subcellular localization, and cellular regulation.

The following are this reviewer’s comments/suggestions/questions regarding the results/interpretations reported and discussed in the manuscript:

Regarding Table 1 and 2, while the color gradient is useful to distinguish the percent deviations from the average, its readability can be improved by inverting the blue color scheme, so the scale will be read from decreasing positive deviations to increasing negative deviations (e., +30%, +20%, +10% à -10%, 20%, -30%). In reference to Figure 1: The authors mentioned that multiple MSF complex types were merged together due to low sample representation. Does this indicate that (i) the number of entries (205 minus 2) in the MFIB database was insufficient; or, (ii) the initial numbers of sequence-based and/or structure-based clusters were excessive? Within the structure clusters 1 and 2, two sequence clusters were merged for each, and all remaining were omitted; however, for structure clusters 3 and 4, all four sequence clusters were grouped. The authors should elaborate on their decision to employ this organization. Furthermore, based on the color classification, it appeared that the structure clusters served as the basis for merging groups to create the five distinctive MSF types. On that note, the authors should clarify their reasoning to group the twenty complexes based on the structure clusters (vertically) rather than the sequence clusters (horizontally). One concern for this approach is that it may be biased by the properties of the final folded/bound structured complexes, and might be neglecting the initial disordered unbound states. (Minor suggestion 1) The black font combined with the dark blue background is difficult to read. Switching to a more contrasting font color (e.g., white) will assist with the readability. (Minor suggestion 2) In the figure’s caption, a brief description of the color scheme might be helpful. Regarding Figure 2, the authors calculated the contribution of interchain/intermolecular interactions to the total energy that stabilizes the complexes. In comparison with Figure 2a, there exist multiple overlaps among the MSF types 2-5 in Figure 2b. The authors should elaborate on the implications of the extensive overlaps. In Figure 3, the author depicted the Kd value distributions for the six types of MSF complexes. Interaction affinities (Kd) highly depend on solution conditions (e.g., temperature, pH, and salt concentrations), and the relationship can be complex and nonlinear. In that case, how can a single Kd value per MSF complex encompass the intricate interactions? (Minor suggestion) The figure’s caption should define the color distribution surrounding the box plots to improve readability. In Figure 4a, regarding the “extrinsic modulation of function” section, what is being modulated if all of the states, independent of methylation, inhibit transcription?

Author Response

We would like to thank the reviewer for the careful reading of the manuscript and the provided positive and constructive comments. The manuscript was revised in light of these comments, and we believe that these corrections have improved the manuscript making the main messages of our work more accessible. After each of the reviewer’s comments we address it in detail (marked in italics).

Regarding Table 1 and 2, while the color gradient is useful to distinguish the percent deviations from the average, its readability can be improved by inverting the blue color scheme, so the scale will be read from decreasing positive deviations to increasing negative deviations (e., +30%, +20%, +10% à -10%, 20%, -30%).

The colour legend to both tables have been changed, now it shows the shades used in the proposed, more logical order, making it clearer that colour depth corresponds to the extent of the deviation from the mean.

In reference to Figure 1: The authors mentioned that multiple MSF complex types were merged together due to low sample representation. Does this indicate that (i) the number of entries (205 minus 2) in the MFIB database was insufficient; or, (ii) the initial numbers of sequence-based and/or structure-based clusters were excessive?

In our view, having a total of 20 possible complex types is reasonable, as the current (rather ad hoc) classification present in MFIB uses 8 classes further divided into 33 subclasses. While these do not constitute a self  consistent classification scheme (as some correspond to a protein family, such as HIUase, while others correspond to a structure type, such as coiled coils), they give some indication of a reasonable number of classes to be defined. Allowing 20 classes fits well to this idea, and hence we opted to keep this number of possible types.

Thus, we believe that the need for merging is mostly due to reason number (i), the low overall number of known MSF examples. Having a larger number of MSF complexes could give an answer for that, but we think that at least some of these lowly populated spaces would have a higher incidence number. Having a larger dataset could provide evidence that currently merged subtypes should be separated, and new independent types should be defined. Our long term goal is to turn MFIB into a regularly maintained database, with the next iteration having a considerably larger number of entries. When this update is complete, we will reassess the classification and will modify it, if necessary.

Within the structure clusters 1 and 2, two sequence clusters were merged for each, and all remaining were omitted; however, for structure clusters 3 and 4, all four sequence clusters were grouped. The authors should elaborate on their decision to employ this organization. 

For structure clusters 1 and 2 the two merged clusters have an apparently higher number of complexes than the ones left out. In the case of structure clusters 3 and 4, all four individual clusters have a comparable number of complexes, so we decided to merge all four of them. We added the following explanation to the corresponding Results section (2.3): “For structure clusters 1 and 2, only two adjacent sequence clusters were merged, as these contain over 95% and 85% of the complexes, respectively. In contrast, for structure classes 3 and 4 all four sequence clusters were merged, as the distribution of complexes is more even across the sequence space. For structure cluster 5, even a single sequence cluster is enough to capture over 85% of complexes, and thus no merging was employed.”

Furthermore, based on the color classification, it appeared that the structure clusters served as the basis for merging groups to create the five distinctive MSF types. On that note, the authors should clarify their reasoning to group the twenty complexes based on the structure clusters (vertically) rather than the sequence clusters (horizontally). One concern for this approach is that it may be biased by the properties of the final folded/bound structured complexes, and might be neglecting the initial disordered unbound states.

We agree that the choice for merging sequence clusters as opposed to structure clusters is non-trivial. The unbound states of the proteins are mainly reflected in the sequence properties. While these are less represented directly when using structure-based clusters are a stronger guide to define types, sequence properties are in fact indirectly reflected in the structural parameters too (for example, structures with a high helical content will naturally be high in helix-promoting residues and low in prolines).

From a more empirical point of view, we found that calculating averages of parameters for groups defined based on structure showed a lower standard deviation that those we tried to define based on sequence. Also, biological parameters, like Kd, energetic properties and subcellular localization, show a better separation for types defined based on structure than for those defined based on sequence.

(Minor suggestion 1) The black font combined with the dark blue background is difficult to read. Switching to a more contrasting font color (e.g., white) will assist with the readability.

We switched the colour for type one complexes in Figure 1 to a lighter shade of the same colour, to make the text more readable.

(Minor suggestion 2) In the figure’s caption, a brief description of the color scheme might be helpful.

We added an explanatory sentence to the figure caption.

Regarding Figure 2, the authors calculated the contribution of interchain/intermolecular interactions to the total energy that stabilizes the complexes. In comparison with Figure 2a, there exist multiple overlaps among the MSF types 2-5 in Figure 2b. The authors should elaborate on the implications of the extensive overlaps.

We added a short section reflecting on this in the appropriate Results section (2.4): “As opposed to the complexes in Figure 2a, MSF complexes show a high overlap in the energy space. This shows that very different structures, with potentially very different sequence compositions can have similar energetic properties. Also, the high variability of energetic properties within complex types (the main reason for high overlap between different groups) shows that depending on the biological function, similar complexes can be required to have very different stabilities. For example, while several dimeric transcription factors can have similar structures that accommodate DNA-binding, the association and dissociation rates of the dimers (regulating their transcriptional activity) have to adapt to the required expression profiles of the genes they regulate.”

In Figure 3, the author depicted the Kd value distributions for the six types of MSF complexes. Interaction affinities (Kd) highly depend on solution conditions (e.g., temperature, pH, and salt concentrations), and the relationship can be complex and nonlinear. In that case, how can a single Kd value per MSF complex encompass the intricate interactions?

We fully agree that true biological Kd values are a complex function of several environmental parameters. Unfortunately, for most MSF complexes we do not have information even about a Kd value in a single environment, or information about the environmental conditions under which they operate, let alone the function that would connect these. The most we could do is to use the default settings of the Kd prediction method used, hoping that individual errors will be masked on average. This is the reason why we rather compare average values instead of drawing conclusions from the shapes of the Kd distributions or the absolute values of their means. We included a sentence in the relevant Results section to highlight this.

(Minor suggestion) The figure’s caption should define the color distribution surrounding the box plots to improve readability.

We realized that the figure’s shading actually contains no information, so we removed it to reduce noise. Now the fill of the violin plot it even grey.

In Figure 4a, regarding the “extrinsic modulation of function” section, what is being modulated if all of the states, independent of methylation, inhibit transcription?

It is true that the resulting biological effect is repression with or without methylation. However, the strength of repression is dependent on the PTM in an indirect way, meaning that the methylation doesn’t affect the MSF complex’s activity, but can enhance it by recruiting an additional protein which switches on a parallel repressor function. We added the following sentence to the corresponding Results section to make this clearer: “This way the strength of repression depends on the PTM of the MSF complex, but through an additional protein that is not part of the complex, but contributes to the complex function through a parallel mechanism in an indirect way.”